# Clustering as Reasoning: A $k$-Means Interpretation of Chain-of-Thought Graph Learning

**Xuanting Xie** [§ * 1 2]  **Zhaochen Guo** [* 1]  **Bingheng Li** [* 3]  **Xingtong Yu** [4]  **Zhifei Liao** [1]  **Zhao Kang** [1 †]  **Yuan Fang** [2 †]

## Abstract

Chain-of-Thought (CoT) prompting has shown promise in enhancing the reasoning capabilities of large language models (LLMs) on text-attributed graphs (TAGs). This work reframes CoT-based graph learning through the principle of clustering as reasoning, offering a $k$-means interpretation of how iterative reasoning operates over graph-structured data. We observe that existing graph CoT methods rely on disjoint architectures and fixed graph representations, limiting step-by-step semantic-topological interaction and interpretability. To overcome this limitation, we propose a unified framework named KCoT that integrates CoT reasoning with graph representation learning. Our key theoretical result reveals a formal mathematical correspondence between a Transformer block and the $k$-means algorithm, allowing reasoning to be interpreted as iterative assignment and update steps. Based on this insight, we introduce a Semantic Discriminating Prompt that explicitly formulates these steps as structured CoT reasoning, together with a structure-grounded alignment strategy to fuse topological priors with evolving thought-conditioned representations. Experiments on standard benchmarks demonstrate consistent improvements over state-of-the-art methods, validating clustering as a principled mechanism for CoT-based graph learning.

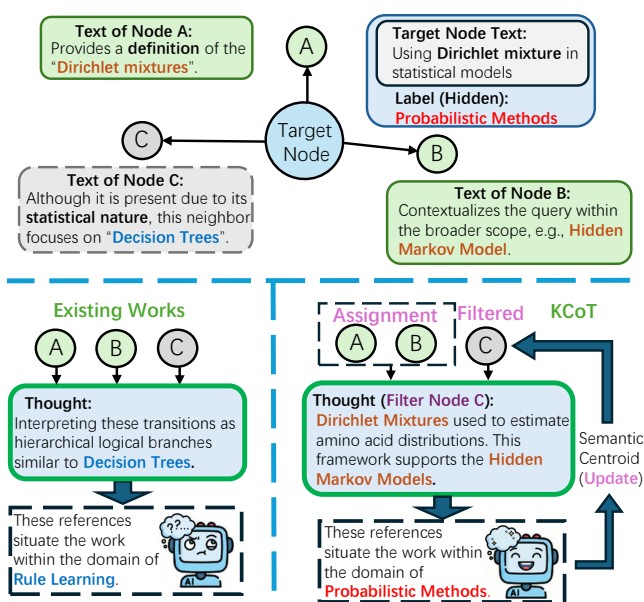

*Figure 1.* A toy example of the proposed prompt on Cora. The model effectively filters irrelevant neighbor C and focuses on identifying the salient semantic features, like "Dirichlet Mixtures" and "Hidden Markov Models".

§Work done while visiting Singapore Management University. *Equal contribution  †Corresponding authors. [1]University of Electronic Science and Technology of China, 611731, Chengdu, China [2]Singapore Management University, 188065, Singapore [3]Michigan State University, East Lansing, 48824, MI, USA [4]The Chinese University of Hong Kong, Hong Kong SAR, China. Correspondence to: Xuanting Xie <x624361380@outlook.com>, Zhao Kang <zkang@uestc.edu.cn>, Yuan Fang <yfang@smu.edu.sg>.

*Proceedings of the 43rd International Conference on Machine Learning*, Seoul, South Korea. PMLR 306, 2026. Copyright 2026 by the author(s).

## 1. Introduction

Graph Chain-of-Thought (CoT) prompting has emerged as a promising paradigm for enhancing the reasoning capabilities of LLMs on Text-Attributed Graphs (TAGs). By decomposing complex problems into intermediate reasoning steps (Wei et al., 2022; Wang et al., 2023b; Chu et al., 2023), CoT allows models to attain expert-level proficiency in sophisticated domains (Feng et al., 2023). Initially, research focused on bridging the modality gap by translating graph topologies into natural language prompts (Jia et al., 2025) or simulating reasoning steps within latent vector spaces for text-free graphs (Yu et al., 2025b). Subsequently, LLMs are fine-tuned with explicit graph reasoning traces to enhance native topological understanding (Luo et al., 2024), and multi-agent systems with tool-chaining have been employed to overcome context limitations and scale reasoning to industrial-sized graphs (Wei et al., 2025; Zhuo et al., 2025a;b). However, despite these achievements, the full

potential of Graph CoT remains largely untapped due to two fundamental challenges, as follows.

First, existing Graph CoT paradigms typically employ disjoint, loosely coupled architectures that separate the processing of LLMs and Graph Neural Networks (GNNs) (Xie et al., 2023; Kang et al., 2024; Xie et al., 2025b) into isolated stages. In these frameworks, LLMs function merely as an independent semantic parser and generator. Thus, the semantic reasoning process operates in a vacuum, detached from the structural propagation mechanism of GNNs (Chen et al., 2024c; Yang et al., 2025d; Li et al., 2025; Xie et al., 2025a). Consequently, this prevents step-by-step semantic-topological interaction, hindering the seamless integration of structural constraints with textual reasoning and leading to suboptimal alignment between the two modalities.

Second, there is limited interpretability of the underlying reasoning mechanism. Current CoT learning often operates as a "black box", lacking geometric interpretability regarding how natural language reasoning drives the optimization of node representations. Existing approaches primarily instruct LLMs to think "step by step" without a clear theoretical grounding (Tang et al., 2024). Thus, it remains unclear how natural language prompting corresponds to a well-defined mathematical objective or how to design mechanisms that explicitly drive the iterative optimization of graph representations. Without such grounding, it is difficult to reconcile the generated thoughts with the graph learning objective.

To address these challenges, we propose the $k$-means Interpretation of Chain-of-Thought Graph Learning (KCoT), a novel framework that frames CoT-based graph learning through the principle of clustering as reasoning. To resolve the interpretability issue, we provide a theoretical analysis demonstrating that a Transformer block in LLMs admits a parameterization that is functionally equivalent to the $k$-means algorithm. Guided by this, we design a *Semantic Discriminating Prompt* that explicitly reformulates the "Assignment" and "Update" steps of $k$-means into CoT reasoning steps. As illustrated in Fig. 1, unlike previous prompts that indiscriminately summarize all neighbors, our prompt acts as a semantic filter to screen unrelated nodes ("Assignment") and distill the semantic centroids ("Update"). To further overcome the semantic-structural misalignment, we propose a structure-grounded thought strategy that synergizes explicit topological priors with induced semantic reasoning. A Condition-Net module is proposed to generate a reasoning matrix, which is conditioned on both graph structures and the evolving reasoning status. We further provide the theoretical analysis demonstrating the advantages of KCoT. Experiments on standard benchmarks demonstrate consistent improvements over state-of-the-art methods, validating our proposed KCoT.

## 2. Related Work

### 2.1. Graph Chain-of-Thought

Building on the CoT paradigm, recent works such as GraphCoT (Jin et al., 2024) integrate the inherent relational structure of TAGs (Yan et al., 2023; Wen & Fang, 2023) to guide LLM reasoning. HetGCoT (Jia et al., 2025) translates graph topologies into natural language prompts, while GCoT (Yu et al., 2025b) simulates reasoning steps within latent vector spaces for text-free graphs. GraphInstruct (Luo et al., 2024) fine-tunes LLMs with explicit graph reasoning traces to enhance topological understanding. Meanwhile, GraphChain (Wei et al., 2025) employs multi-agent systems and tool-chaining to overcome context limitations, scaling reasoning to industrial-sized graphs. GraphGPT (Tang et al., 2024) aligns text with structural data, enabling the model to perform step-by-step reasoning. However, these methods treat CoT as a "black box", lacking interpretability regarding its underlying mechanisms.

### 2.2. Large Language Models for Graphs

Driven by the rapid evolution and generalization capabilities of Large Language Models (LLMs), there is significant interest in leveraging them to address transferability challenges in graph machine learning (Guo et al., 2023; He et al., 2025). Initial attempts focused on linearizing graph structures into textual descriptions (Chen et al., 2024b; Wang et al., 2023a; Liu & Wu, 2023). However, this approach often yields suboptimal performance due to the loss of structural information (Huang et al., 2023). Alternative paradigms utilize LLMs merely as feature enhancers (Xia et al., 2024; Ye et al., 2024) to augment node attributes or synthesize pseudo-labels. Because these frameworks rely on GNNs as the final predictor, they inherit the generalization limitations of GNNs, restricting cross-domain transferability. Consequently, recent research has pivoted towards utilizing LLMs as standalone graph predictors. For example, LLaGA (Chen et al., 2024a) introduces a projector-based encoding scheme to map graph structural data into token sequences compatible with the LLM embedding space. However, these methods often employ disjoint architectures that separate LLM and GNN processing into isolated stages. Consequently, this separation limits semantic–topological interaction, leading to suboptimal performance.

## 3. Preliminaries

**Text-Attributed Graphs.** Formally, we define a graph as $\mathcal{G} = (\mathcal{V}, \mathcal{E}, \mathcal{X})$, where $\mathcal{V}$ and $\mathcal{E}$ represent the sets of nodes and edges, respectively. The edge set $\mathcal{E}$ captures the structural dependencies among the entities in $\mathcal{V}$. Additionally, $\mathcal{X}$ denotes the node feature set, where each node $v_i \in \mathcal{V}$ is associated with a specific feature embedding $\mathbf{X}_i \in \mathbb{R}^d$. This

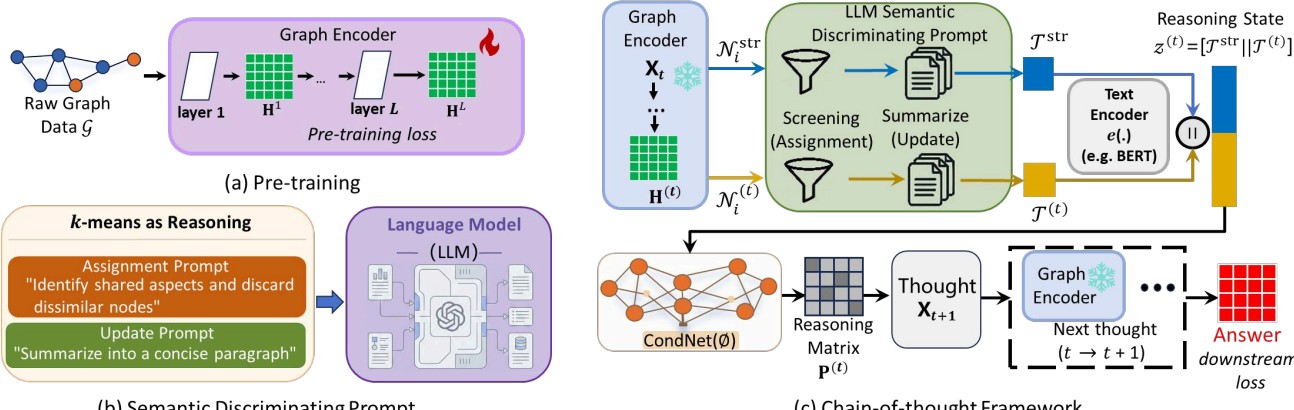

**Figure 2.** The overall framework of KCoT. It synergizes iterative CoT reasoning with graph representation learning. Specifically, we design a Semantic Discriminating Prompt that reformulates the $k$-means into explicit reasoning to refine node semantics. Simultaneously, a Structure-grounded Thoughts Construction strategy and a Condition-Net module dynamically fuse these evolving semantic thoughts with fixed topological priors to achieve semantic-structural alignment.

work specifically targets text-attributed graphs, wherein the attribute $\mathbf{X}_i$ consists of raw textual information $\mathbf{T}_i$ rather than simple numerical features.

**Graph Encoder.** GNNs have established themselves as the predominant encoder architecture. Formally, let $\mathbf{H}^l$ denote the node embedding matrix at the $l$-th layer, where the $i$-th row $\mathbf{h}_i^l$ corresponds to the embedding of node $v_i$. The $L$ layer GNN update is defined as (Zhuo et al., 2023; Yang et al., 2025b;a):

$$\mathbf{H}^L = \text{GRAPHENCODER}(\mathbf{X}, \mathcal{G}; \Theta), \qquad (1)$$

where $\Theta = (\theta^1, \ldots, \theta^L)$ denotes the weights of the these $L$ layers. For brevity, the final output $\mathbf{H}^L$ is referred to as $\mathbf{H}$.

**Pre-training.** Recent studies (Yu et al., 2024; Yang et al., 2025c; Fang et al., 2022; 2025a; 2026) demonstrate that mainstream contrastive pre-training tasks on graphs (Liu et al., 2023; Zhuo et al., 2024a;c; Fang et al., 2025b;c) can be unified under a generalized similarity calculation framework. Formally, we define the unified pre-training objective $\mathcal{L}(\Theta)$ as:

$$\mathcal{L}(\Theta) = -\sum_{o \in \mathcal{T}_{\text{pre}}} \ln \frac{\sum_{a \in \mathcal{P}_o} \exp(\text{sim}(\mathbf{h}_a, \mathbf{h}_o)/\tau)}{\sum_{b \in \mathcal{N}_o} \exp(\text{sim}(\mathbf{h}_b, \mathbf{h}_o)/\tau)}, \quad (2)$$

where $\mathcal{P}_o$ and $\mathcal{N}_o$ denote the sets of positive and negative samples for a target instance $o$, respectively. $\Theta$ denotes the learnable parameters. The vectors $\mathbf{h}_o$, $\mathbf{h}_a$, and $\mathbf{h}_b$ represent the embeddings of the target, positive, and negative instances, respectively. The hyperparameter $\tau$ controls the temperature scaling of the similarity function $\text{sim}(\cdot, \cdot)$. Following established paradigms (Zhuo et al., 2024b; Yu et al., 2024), we adopt link prediction as the specific pre-training pretext task within this similarity-based framework.

# 4. Chain-of-Thought Graph Learning

In this section, we present our framework, which is summarized in Fig. 2.

## 4.1. Motivation

While LLMs demonstrate superior semantic understanding, they often suffer from a lack of interpretability regarding the mechanisms underlying their effectiveness. To bridge this gap, we look beyond the standard view of LLMs as static encoders. We propose that the self-attention mechanism—the backbone of LLMs—can mathematically implement the optimization dynamics of clustering algorithms. Specifically, we demonstrate that a Transformer block can approximate the Assignment and Update steps of the $k$-means algorithm (Xie et al., 2026b;a; Guo et al., 2025; Hou et al., 2025), which relies solely on weight and bias constructions, without modifying or augmenting the input representations.

**Proposition 4.1** (Transformer as $k$-Means Clustering Mechanism). *For any input representations, there exists a parameterization of a Transformer self-attention layer such that its attention weights form an $\epsilon$-approximation of soft $k$-means clustering assignments. Moreover, under a specific construction, this approximation becomes exact with $\epsilon = 0$. As a result, stacking such self-attention layers induces an iterative assignment–update process that mirrors the dynamics of soft clustering methods.*

The proof can be found in A.1.

*Remark* 4.2 (Prompting as a Generalization of Clustering in Reasoning). Proposition 4.1 suggests that self-attention naturally implements a clustering-like assignment–update mechanism. Prompt-based multi-step reasoning can be viewed as a generalization of this process, where intermediate thoughts provide contextualized updates and stabilize

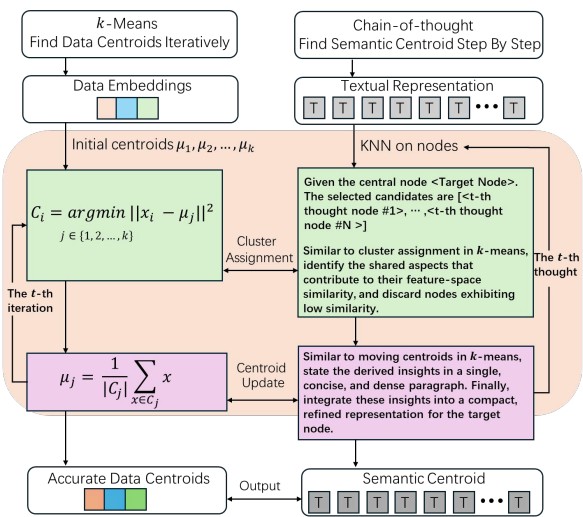

*Figure 3.* LLM-driven replication of the $k$-means. We achieve fine-grained alignment between the $k$-means algorithm and the proposed prompt.

evolving semantic clusters. This enables iterative refinement of latent groupings through language.

Moreover, prior work (Diaz-Rodriguez, 2025) has revealed that LLMs possess a superior capability to extract semantic centroids compared to traditional $k$-means algorithms, which is verified using the $dist$ metric (i.e., the Euclidean distance between the learned and ground-truth centroids). Motivated by these insights, we design our CoT prompts to emulate the Assignment and Update steps of $k$-means. By forcing the model to articulate this iterative process, we enable it to leverage its superior semantic centroid finding ability to progressively generate more accurate thoughts for TAGs.

### 4.2. Semantic Discriminating Prompt

While LLMs have demonstrated remarkable capabilities in semantic understanding, they lack an intrinsic mechanism to perform the rigorous, iterative state updates characteristic. Current approaches treat the clustering algorithm and the language model as separate modules, preventing continuous optimization in the semantic space. The primary objective of this section is to resolve this disjointedness by investigating how the Assignment and Update phases of $k$-means can be reformulated as explicit CoT reasoning steps, thereby grounding the optimization process directly within the LLM's generative framework.

**Prompt Formulation.** The rationale behind this design is to address the semantic misalignment often found in GNNs. In standard $k$-means (left side of Fig. 3), a node $x_i$ is assigned to a cluster $C_j$ by minimizing the Euclidean distance $\|x_i - \mu_j\|^2$. However, in TAGs, "distance" is often subjec-

tive and context-dependent. Therefore, we replace the rigid mathematical distance with LLM-driven discriminative reasoning.

**The "Assignment" Prompt:** We instruct the LLM to act as a semantic filter. Instead of blindly summarizing all neighbors in previous LLM-based methods, the prompt explicitly asks the model to "identify shared aspects" and "discard nodes exhibiting low similarity". This mimics the $k$-means assignment step by calculating the "semantic distance" between the target node and its candidate neighbors, effectively ensuring that the retained neighboring information is rigorously aligned with the underlying semantic context.

**The "Update" Prompt:** In $k$-means, the centroid $\mu_j$ is updated by averaging the vectors in the cluster. In our framework, we cannot mathematically average text. Instead, we design the prompt to perform abstractive summarization, asking the model to "state derived insights in a single, concise, and dense paragraph". This forces the LLM to compress the semantic variance of the selected neighbors into a "compact refined representation", which mathematically serves as the updated "Semantic Centroid" for the thought.

Finally, the output of the text prompt for node $v_i$ is defined as:

$$\mathcal{T}_i \leftarrow \text{Prompt}(\mathbf{T}_i, \mathbf{N}_i), \qquad (3)$$

where $\mathbf{N}_i$ indicates the selected candidates of node $v_i$. All prompt designs are summarized in the right side of Fig. 3.

### 4.3. Thought Construction

Next, we introduce the inference framework and prompts designed for the text information in our architecture.

During the $t$-th inference iteration, we input the query graph $\mathcal{G} = (\mathcal{V}, \mathcal{E}, \mathcal{X})$, alongside its thought-based feature matrix, into the pre-trained encoder:

$$\mathbf{H}^{(t)} = \text{GRAPHENCODER}(\mathbf{X}_t, \mathbf{G}; \Theta_0), \qquad (4)$$

where $\Theta_0$ represents the frozen parameters of the pre-trained graph encoder, while $\mathbf{X}_t$ denotes the node feature matrix derived from the preceding $(t-1)$-th thought. The specific mechanisms for prompt generation and feature adaptation are detailed in the subsequent section. It is important to note that for the initial thought ($t = 1$), the raw features are employed directly, such that $\mathbf{X}_1 = \mathbf{X}$.

**Structure-grounded Thought Construction.** To bridge the gap between the integration of graph structures and LLMs, we propose a structure-grounded thoughts strategy, which integrates both fixed topological structures and evolving reasoning. First, we randomly select $K$ neighbors as $\mathcal{N}_i^{\text{str}}$ based on the local graph topology (both 1- and 2-hop

neighbors), ensuring the retention of explicit geometric priors. Simultaneously, to capture the current thought status, we identify the reasoning-induced neighbors $\mathcal{N}_i^{(t)}$ by performing $K$-Nearest Neighbor ($KNN$) search on the node representations $\mathbf{H}^{(t)}$. Based on the Semantic Discriminating Prompt, the obtained textual representations are:

$$
\begin{aligned}
\mathcal{T}_i^{\text{str}} &\leftarrow \text{Prompt}\left(\mathbf{T_i}, \{\mathbf{T_j}, v_j \in \mathcal{N}_i^{\text{str}}\}\right) \\
\mathcal{T}_i^{(t)} &\leftarrow \text{Prompt}\left(\mathbf{T_i}, \{\mathbf{T_j}, v_j \in \mathcal{N}_i^{(t)}\}\right).
\end{aligned} \quad (5)
$$

Given a text encoder $e(\cdot)$, in our case a BERT model (Devlin et al., 2019), the output reasoning state $z^{(t)}$ is formulated as:

$$
\begin{aligned}
T^{\text{str}} &= e(\mathcal{T}^{\text{str}}) \\
T^{(t)} &= e(\mathcal{T}^{(t)}) \\
z^{(t)} &= [T^{str} \| T^{(t)}]
\end{aligned} \quad (6)
$$

### 4.4. Thought-conditioned Learning

We utilize a conditional network (i.e., condition-net) (Yu et al., 2025a) to synthesize $T^{\text{str}}$ and $T^{(t)}$, balancing them in a unified space.

Specifically, for condition-net on the status $z^{(t)}$, the COND-NET generates a reasoning matrix $\mathbf{P}^{(t)} \in \mathbb{R}^{|\mathcal{V}| \times d}$, formulated as:

$$
\mathbf{P}^{(t)} = \text{CONDNET}(z^{(t)}; \phi), \quad (7)
$$

where CONDNET acts as a lightweight adapter (we implement as a Multilayer Perceptron parameterized by $\phi$), functioning as a lightweight hypernetwork (Ha et al., 2022). On one hand, it bridges the gap by transforming linguistic semantics into vector embeddings that are compatible with the graph representation space. On the other hand, it balances the trade-off between fixed topological connectivity and evolving thoughts.

This reasoning matrix is then employed to modulate the query graph's node features for the inference in the next step:

$$
\mathbf{X}_{t+1} = \mathbf{P}^{(t)} \odot \mathbf{X}, \quad (8)
$$

where $\odot$ denotes the element-wise product. $\mathbf{X}_{t+1}$ serves as the input for the pre-trained graph encoder in the $(t+1)$-th iteration. We call the output of the last reasoning step the answer matrix, since it is directly used for downstream tasks to produce final predictions.

### 4.5. Downstream Loss Function

We employ task-specific objective functions to optimize the model. Let $y$ denote the ground-truth labels and $\hat{y}$ represent the predicted probabilities derived from the final node representations.

**Node Classification.** For multi-class node classification, we minimize the Cross-Entropy (CE) loss over the training nodes $\mathcal{V}_{train}$:

$$
\mathcal{L}_{\text{NC}} = - \sum_{i \in \mathcal{V}_{train}} \sum_{c=1}^{C} y_{ic} \log(\hat{y}_{ic}), \quad (9)
$$

where $C$ is the number of classes, and $\hat{y}_{ic}$ is the predicted probability that node $i$ belongs to class $c$.

**Link Prediction.** For link prediction, we minimize the Binary Cross-Entropy (BCE) loss:

$$
\mathcal{L}_{\text{LP}} = - \sum_{(i,j) \in \mathcal{E}_{train}} \left[y_{ij} \log \hat{y}_{ij} + (1 - y_{ij}) \log(1 - \hat{y}_{ij})\right], \quad (10)
$$

where $y_{ij}$ indicates the existence of an edge between node $i$ and $j$, and $\hat{y}_{ij}$ is the predicted probability.

### 4.6. Algorithm and Complexity Analysis

We summarize the KCoT procedure in Algorithm 1. Lines 1–3 initialize the node representations via the graph encoder and construct structural thoughts from the text prompt and graph topology, which are then embedded into continuous representations. Lines 5–13 implement $M$ iterative inference steps: at each step, the model re-encodes node features, retrieves relevant neighbors via KNN, and constructs step-specific thoughts, from which a conditional gating matrix $\mathbf{P}^{(t)}$ modulates the node features for the next iteration.

---

**Algorithm 1** KCoT

---

**Require:** Graph $\mathcal{G} = (\mathcal{V}, \mathcal{E}, \mathbf{X})$, text prompt $\boldsymbol{p}$, steps $M$
**Ensure:** Predicted answer $\hat{y}$
1: $\mathbf{H}^{(0)} \leftarrow \text{GraphEncoder}(\mathbf{X}, \mathcal{G})$
2: $\boldsymbol{T}^{\text{str}} = \text{ThoughtsConstruction}(\boldsymbol{p}, \mathcal{G})$
3: $T^{\text{str}} = e(\boldsymbol{T}^{\text{str}})$
4: /* KCoT inference steps */
5: **for** $t = 1$ to $M$ **do**
6:     $\mathbf{H}^{(t-1)} \leftarrow \text{GraphEncoder}(\mathbf{X}^{(t-1)}, \mathcal{G})$
7:     KNN on $\mathbf{H}^{(t-1)}$
8:     $\boldsymbol{T}^t \leftarrow \text{ThoughtsConstruction}(\boldsymbol{p}, \text{KNN})$
9:     $T^{(t)} = e(\boldsymbol{T}^{\text{str}})$
10:    $z^{(t)} = [T^{\text{str}} \| T^{(t)}]$
11:    $\mathbf{P}^{(t)} = \text{CondNet}(z^{(t)}; \phi)$
12:    $\mathbf{X}^{(t)} \leftarrow \mathbf{X}^{(t-1)} \odot \mathbf{P}^{(t)}$
13: **end for**
14: $\hat{y} = \text{Downstream}(\mathbf{H}^{(M)})$
15: **Return** $\hat{y}$

---

The complexity of KCoT lies in a base GNN with $d_L$ hidden dimensions per layer, identifying reasoning-induced neighbors through KNN, and thought generation with an

LLM. Assuming a bounded prompt length and a constant inference cost $C_{\text{LLM}}$ per call (dependent on the specific LLM backbone), the total time complexity for each reasoning iteration $t$ is the summation of these components:

$$\mathcal{O}_{\text{total}} = \mathcal{O}(\underbrace{L(|\mathcal{V}| + |\mathcal{E}|)d_L}_{\text{GNN}} + \underbrace{|\mathcal{V}|d_L}_{\text{KNN}} + \underbrace{|\mathcal{V}|C_{\text{LLM}}}_{\text{LLM}})$$

### 4.7. Theoretical Analysis: Semantic–Structural Alignment

We now provide a theoretical perspective on how KCoT reasoning contributes to the alignment of semantic and structural information in our framework. Our analysis is based on the following observation: while message passing in GNNs relies on structural neighborhoods, LLM-based semantic reasoning introduces an independent perspective based on representation similarity. The effectiveness of CoT thus hinges on aligning these two views.

**Semantic–Structural Misalignment.** To quantify the interaction between structural and semantic information, we introduce a label-free misalignment metric.

**Definition 4.3** (Semantic–Structural Misalignment). Let $\mathcal{N}_i^{\text{str}}$ and $\mathcal{N}_i^{(t)}$ in Eq. (5) denote the structural and semantic neighborhoods of node $i$. Define the per-node misalignment as

$$\delta_i^{(t)} := 1 - \frac{|\mathcal{N}_i^{\text{str}} \cap \mathcal{N}_i^{(t)}|}{|\mathcal{N}_i^{\text{str}} \cup \mathcal{N}_i^{(t)}|},$$

and the global misalignment score as

$$\Delta_t := \mathbb{E}_{i \sim V}[\delta_i^{(t)}].$$

This metric captures how much the semantic similarity is inconsistent with the structural prior. Smaller $\Delta_t$ indicates better semantic–structural alignment.

Under mild assumptions A.6 and A.7, we can formally show that CoT reduces misalignment over iterations:

**Theorem 4.4** (CoT Contracts Semantic–Structural Misalignment). *There exist constants $0 < \rho < 1$ and $\varepsilon \geq 0$ such that*

$$\Delta_{t+1} \leq \rho \Delta_t + \varepsilon \quad \text{for all } t \in \mathbb{Z}^+.$$

*In particular, when $\varepsilon$ is small, the misalignment decreases geometrically until it reaches an error floor $\mathcal{O}(\varepsilon)$.*

Although we do not assume access to labels, the misalignment score $\Delta_t$ provides a meaningful proxy for representation quality. When $\Delta_t$ is high, message passing aggregates semantically inconsistent neighbors, leading to representation blur and class mixing. As $\Delta_t$ decreases, message passing becomes more semantically coherent, improving representation separability.

*Remark* 4.5 (Role of CoT). This analysis suggests that CoT is best viewed not as "smarter reasoning," but as an iterative alignment mechanism between LLM-based semantics and GNN-based structure. By reducing semantic–structural misalignment, it stabilizes message passing and limits cross-class contamination.

Combining the clustering interpretation of Transformer-based reasoning (Proposition 4.1) with the semantic–structural contraction property of CoT inference (Theorem 4.4), we obtain the following corollary.

**Corollary 4.6.** *Under the same setting as Theorem 4.4 with a frozen graph encoder, our CoT reasoning framework admits a stronger mechanism for aligning semantic and structural information, compared to classic $k$-means clustering.*

Classic $k$-means assignment–update dynamics operate solely in the representation space and improve semantic compactness, but they do not explicitly act on the semantic–structural misalignment $\Delta_t$, which depends on the structural neighborhood $\mathcal{N}_i^{\text{str}}$. In contrast, Theorem 4.4 establishes that CoT iterations satisfy $\Delta_{t+1} \leq \rho \Delta_t + \varepsilon$, for some $0 < \rho < 1$. This contraction property implies that CoT explicitly reduces semantic–structural misalignment across iterations, a mechanism absent from $k$-means clustering. Consequently, CoT offers a more effective alignment between semantic and structural information in downstream graph learning.

*Table 1.* Dataset Statistics

| Dataset | Domain | #Node | #Edge | #Sparsity(‰) |
|---------|--------|-------|-------|--------------|
| Cora | citation | 2708 | 5429 | 14.8065 |
| Pubmed | citation | 19717 | 44338 | 2.2810 |
| Arxiv | citation | 169343 | 1166243 | 0.8134 |
| Products | e-commerce | 2449029 | 61859140 | 0.2063 |

## 5. Experiments

In this section, we conduct experiments to evaluate KCoT, and analyze the empirical results.

### 5.1. Experimental Setup

**Datasets.** To verify the effectiveness of our proposed model, we conduct experiments on several widely used text-attributed graph datasets including Pubmed, Cora (Yang et al., 2016), ogbn-Arxiv, and ogbn-Products (Hu et al., 2020). Spanning citation networks and e-commerce domains, these datasets exhibit distinct structural properties. Ranging from small-scale to massive, they provide a rigorous testing ground for models in both sparse and dense scenarios. Dataset statistics are summarized in Table 1.

**Baselines.** We compare our model against various state-of-

*Table 2.* Performance comparison with baseline models under three evaluation protocols. **Single Focus** represents models trained on an individual task-dataset pair. **Task Expert** refers to models specialized in one specific task by training across all datasets. **Classification Expert** denotes models jointly trained on node classification and link prediction across all datasets to achieve cross-domain proficiency in classification. The **best** results are highlighted in bold, and the second best results are underlined. We report the statistical significance over the strongest baseline using the $t$-test.

| MODEL TYPE | MODEL | NODE CLASSIFICATION ACCURACY (%) | | | | LINK PREDICTION ACCURACY (%) | | | |
|---|---|---|---|---|---|---|---|---|---|
| | | ARXIV | PRODUCTS | PUBMED | CORA | ARXIV | PRODUCTS | PUBMED | CORA |
| SINGLE FOCUS | GCN | 73.72 | 80.75 | 92.96 | 88.93 | 91.43 | 93.95 | 90.91 | 81.59 |
| | GraphSAGE | 76.29 | 82.87 | 94.87 | 88.89 | 91.64 | 94.96 | 90.64 | 79.15 |
| | GAT | 74.06 | 83.06 | 92.33 | 88.97 | 85.99 | 93.85 | 83.96 | 80.06 |
| | SGC | 71.77 | 75.47 | 87.35 | 87.97 | 87.99 | 88.51 | 83.60 | 80.94 |
| | SAGN | 75.70 | 82.58 | 95.17 | 89.19 | 90.62 | 94.85 | 90.48 | 79.88 |
| | NodeFormer | 74.85 | 83.72 | 94.90 | 88.23 | 91.84 | 90.93 | 77.69 | 77.26 |
| | Vicuna-7B | 49.62 | 55.63 | 60.15 | 55.43 | 80.23 | 86.77 | 73.25 | 78.45 |
| | GraphGPT | 75.11 | 84.15 | 94.23 | 88.45 | 91.28 | 94.32 | 82.50 | 80.19 |
| | LLAGA-ND | 75.98 | 84.60 | 95.03 | 88.86 | 91.24 | **97.36** | 91.41 | 83.79 |
| | LLAGA-HO | 76.66 | 84.67 | 95.03 | 89.22 | 94.15 | 95.56 | 89.18 | 86.82 |
| | KCoT | **79.25** | **86.39** | **95.87** | **90.63** | **95.14** | 96.70 | **93.08** | **88.45** |
| | *p*-value | 7.96e-8 | 1.52e-6 | 6.27e-5 | 7.34e-6 | 8.25e-5 | 8.33e-3 | 2.21e-5 | 2.66e-5 |
| TASK EXPERT | GCN | 71.45 | 80.88 | 89.25 | 81.62 | 88.51 | 93.54 | 81.01 | 78.88 |
| | GraphSAGE | 72.56 | 82.50 | 94.15 | 81.99 | 87.76 | 93.49 | 76.14 | 80.74 |
| | GAT | 72.19 | 82.61 | 87.97 | 83.58 | 82.58 | 92.03 | 76.85 | 79.76 |
| | NodeFormer | 72.35 | 82.99 | 94.41 | 83.27 | 84.11 | 93.42 | 80.40 | 81.03 |
| | Vicuna-7B | 48.72 | 65.25 | 67.87 | 54.66 | 80.36 | 85.64 | 79.43 | 80.28 |
| | GraphGPT | 73.50 | 84.32 | 94.12 | 88.95 | 90.82 | 93.12 | 83.40 | 81.27 |
| | LLAGA-ND | 76.41 | 84.60 | 94.78 | 88.19 | 91.20 | **97.38** | 93.27 | 89.41 |
| | LLAGA-HO | 76.40 | 84.18 | 95.06 | 89.85 | 94.36 | 95.85 | 88.88 | 87.50 |
| | KCoT | **79.26** | **86.80** | **96.79** | **91.64** | **95.72** | 96.32 | **95.97** | **91.34** |
| | *p*-value | 3.72e-8 | 2.58e-6 | 1.34e-5 | 1.60e-5 | 9.52e-5 | 5.29e-4 | 5.46e-7 | 7.32e-6 |
| CLASSIFICATION EXPERT | GCN | 70.95 | 80.02 | 89.00 | 82.77 | 87.69 | 92.88 | 72.28 | 78.35 |
| | GraphSAGE | 71.91 | 81.62 | 91.81 | 82.44 | 89.23 | 92.22 | 75.36 | 82.09 |
| | GAT | 70.90 | 81.83 | 87.72 | 82.07 | 85.18 | 92.11 | 75.00 | 80.35 |
| | NodeFormer | 63.20 | 75.55 | 89.50 | 69.19 | 82.33 | 75.42 | 78.22 | 81.47 |
| | Vicuna-7B | 55.26 | 67.69 | 70.91 | 68.62 | 81.34 | 86.58 | 78.09 | 81.04 |
| | GraphGPT | 68.74 | 82.13 | 93.88 | 87.60 | 91.58 | 94.26 | 82.46 | 83.50 |
| | LLAGA-ND | 75.85 | 83.58 | 95.06 | 87.64 | 90.81 | 96.56 | 92.36 | 87.35 |
| | LLAGA-HO | 75.99 | 83.32 | 94.80 | 89.30 | 94.30 | 96.06 | 88.64 | 88.53 |
| | KCoT | **78.76** | **86.25** | **96.27** | **91.34** | **95.45** | **97.30** | **93.40** | **90.34** |
| | *p*-value | 4.83e-8 | 6.49e-7 | 5.50e-5 | 4.92e-6 | 3.02e-4 | 4.54e-3 | 5.37e-5 | 1.68e-6 |

*Table 3.* Ablation study on the effects of key components (Accuracy%).

| VARIANT | NODE CLASSIFICATION | | LINK PREDICTION | |
|---|---|---|---|---|
| | CORA | PRODUCTS | CORA | PRODUCTS |
| KCoT w/o $\mathcal{N}^{str}$ | 89.84 | 85.12 | 87.68 | 96.03 |
| KCoT w/o $\mathcal{N}^{(t)}$ | 89.02 | 84.17 | 85.32 | 94.47 |
| KCoT w/o Prompt | 87.97 | 82.35 | 83.47 | 92.05 |
| KCoT w/o CoT | 89.12 | 82.47 | 82.65 | 94.21 |
| KCoT | **90.63** | **86.39** | **88.45** | **96.70** |

the-art methods. The first category includes GNNs: GCN (Kipf & Welling, 2017), GraphSage (Hamilton et al., 2017), GAT (Velickovic et al., 2017), SGC (Wu et al., 2019) , and SAGN (Sun et al., 2025). The second category consists of the graph transformer NodeFormer (Wu et al., 2022). Finally, we include the open-source LLM Vicuna-7B as a base-line for text-attributed graph understanding, alongside recently proposed LLM-based approaches such as GraphGPT (Tang et al., 2024) and LLAGA (Chen et al., 2024a).

**Implementation Details.**[1] To ensure fair comparison, we adopt the experimental settings of LLAGA (Chen et al., 2024a). Specifically, we adhere to the standard partition ratios: 6:2:3 for Arxiv, 8:2:90 for Products, and 6:2:2 for both Cora and Pubmed. For link prediction, we employ their sampling strategy by deriving node pairs from the corresponding node-level subsets, ensuring the edge-level training size matches that of the node-level task. Our framework integrates Vicuna-7B (v1.5, 16K) to serve as the underlying backbone. A 2-layer GCN with a hidden dimension of 128 serves as the graph encoder for all datasets. Hidden dimen-

---

[1]Code is available at https://github.com/Uncnbb/KCoT.

sion $d$ in the condition-net is selected from $\{16, 32, 64, 128\}$. The number of thought steps $t$ and neighbor count $K$ are fixed at 2 and 5, respectively. We set pre-training epochs to 1000 and downstream training epochs to 400, updating thoughts every 100 epochs for efficiency. We implemented an early-stopping mechanism during the optimization phase. Training was terminated prematurely if the validation performance failed to improve for several successive epochs. We evaluate performance on node classification and link prediction, using Accuracy as the primary metric. Experiments were conducted on a system with 4 NVIDIA A800 (80GB) GPUs, 14 Intel Xeon Gold 6348 CPU cores, and 100GB of RAM.

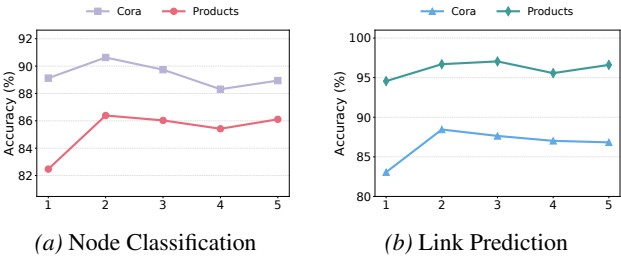

*(a)* Node Classification       *(b)* Link Prediction

*Figure 4.* Impact of Thought Length $t$.

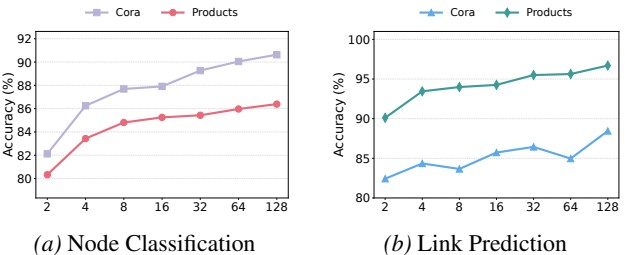

*(a)* Node Classification       *(b)* Link Prediction

*Figure 5.* Impact of hidden dimension $d$ in the condition-net.

### 5.2. Overall Performance Comparison

Table 2 compares our proposed framework and state-of-the-art baselines across three evaluation protocols. KCoT achieves the best performance in most cases, demonstrating a clear advantage, for example, reaching 79.25% accuracy on OGBN-ARXIV. The core superiority of our model lies in its multi-step CoT reasoning, which provides a decisive advantage over existing baselines. Unlike general-purpose LLMs such as Vicuna-7B, which rely exclusively on textual semantics and lack structural awareness, our approach explicitly leverages graph structures. Compared to GraphGPT which also incorporates CoT, our model features a superior interpretable design, providing more effective step-by-step reasoning paths. In contrast to LLAGA's template-based structural serialization, our method achieves a progressive alignment between textual semantics and structural information. This alignment approximates $k$-means clustering,

where each CoT step functions as a latent centroid update to refine decision boundaries and filter structural noise. This yields geometric interpretability significantly superior to the post-hoc explanations of prior models. Furthermore, the poor performance of traditional methods like GCN and GraphSAGE highlights the significance of leveraging LLM capabilities.

*Table 4.* Integration with Various LLMs (Accuracy%)

| MODEL | NODE CLASSIFICATION | | LINK PREDICTION | |
|---|---|---|---|---|
| | CORA | PRODUCTS | CORA | PRODUCTS |
| Vicuna-7B | 90.63 | 86.39 | 88.45 | 96.70 |
| Llama2-7B | 89.92 | 87.17 | 88.82 | 96.93 |
| ChatGPT 4-1 nano | 91.04 | 87.85 | 89.22 | 97.81 |

### 5.3. Ablation Study

To systematically validate the contribution of each component in KCoT, we conduct an ablation study under the SINGLE FOCUS protocol. Table 3 presents a comparative analysis between the full model and four distinct variants. (1) **KCoT w/o $\mathcal{N}^{\text{str}}$** discards inputting graph topology into LLM, exhibits a performance decline. This indicates that the structure-grounded design enhances the interaction between the LLM and graph structures. (2) Removing the KNN-based neighbors (**KCoT w/o $\mathcal{N}^{(t)}$**) leads to a further drop in accuracy. This confirms that relying solely on fixed graph edges is insufficient, especially for nodes with sparse or noisy connections. (3) Most notably, the removal of our specific prompt mechanism (**KCoT w/o Prompt**, i.e., inputting only original text), results in the worst performance in most cases. This verifies that the LLM requires explicit algorithmic guidance, rather than acting merely as a direct text encoder. (4) The variant **KCoT w/o CoT** restricts the inference process to a single pass (i.e., $t = 1$), disabling the iterative refinement mechanism. This also leads to a consistent degradation across all tasks, showing that CoT offers a more effective coordination between semantic and graph structure. This conclusion is consistent with previous theoretical analysis.

### 5.4. Sensitivity Analysis

We examine the sensitivity of two key hyperparameters: the length of thought ($t$) and the hidden dimension of the condition-net ($d$). Figure 4 illustrates model performance across varying thought lengths. We observe a consistent pattern where performance generally peaks at $t = 2$ across different tasks. Beyond this point ($t > 2$), performance tends to decline. On one hand, this behavior mirrors $k$-means clustering, where excessive iterations can overfit noise by pulling centroids toward outliers. On the other hand, since our reasoning mechanism is GNN-based, stacking too many layers may lead to over-smoothing, rendering KNN neighbor selection inaccurate. Additionally, Figure 5 shows the

impact of varying the condition-net hidden dimension $d$. Increasing $d$ consistently improves performance, indicating that higher-dimensional projections enhance expressivity and facilitate better representation learning.

### 5.5. Integration with Various LLMs

Our framework demonstrates remarkable flexibility across various base LLMs. As shown in Table 4, we evaluate performance across Vicuna-7B, Llama2-7B, and the more advanced ChatGPT-4.1 nano. It is evident that our method consistently yields favorable results irrespective of the base LLM, verifying its robustness across different architectures. Furthermore, the integration with ChatGPT-4.1 nano achieves the highest accuracy, indicating that our approach effectively leverages the superior reasoning capabilities of stronger backbones to further enhance the performance.

### 5.6. Visualization

To intuitively evaluate representation quality and support our theoretical analysis, we employ t-SNE to visualize node embeddings and track the inter/intra-class ratio on the CORA dataset. Initially, raw features (Fig. 6a) exhibit a disordered distribution with significant class overlap. By Epoch 200 (Fig. 6c) and Epoch 400 (Fig. 6e), distinct clusters emerge; samples gravitate towards class centers, and decision boundaries sharpen. Furthermore, Fig. 6f illustrates a rising trend in the average inter/intra-class ratio, indicating continuous improvement in cluster discriminability. This aligns with our theoretical framework, which interprets CoT reasoning as an iterative assignment and update procedure analogous to $k$-means. These results also empirically verify our definitions and assumptions for our main Theorem. The Traditional Prompt (Fig. 6b), which relies on a learnable matrix without LLM-based text information, fails to achieve distinct separation, highlighting the necessity of semantic knowledge. Finally, regarding the effectiveness of CoT, we visualize the single-step ablation ($t = 1$) in Fig. 6d. Compared to $t = 2$ (Fig. 6e), the clusters at $t = 1$ are formed but less distinct, suggesting that a single inference step struggles with complex boundaries. This confirms that our multi-step CoT mirrors iterative $k$-means to refine latent centroids and correct misclassifications, achieving superior class separation.

## 6. Conclusion

In this work, we presented KCoT, a unified framework that reframes CoT-based graph learning through the principle of clustering as reasoning. By establishing a theoretical isomorphism between Transformer blocks and $k$-means, we bridge the gap between linguistic generation and geometric optimization. Our approach utilizes a Semantic Discriminating Prompt to explicitly formulate reasoning as cluster

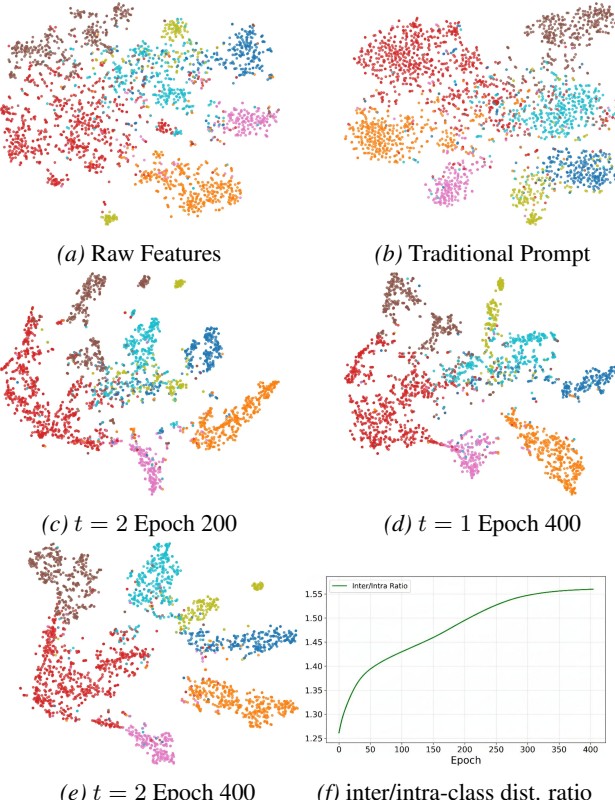

*(a)* Raw Features     *(b)* Traditional Prompt

*(c)* $t = 2$ Epoch 200     *(d)* $t = 1$ Epoch 400

*(e)* $t = 2$ Epoch 400     *(f)* inter/intra-class dist. ratio

*Figure 6.* t-SNE visualization of node embeddings (different colors represent different classes) and evolution of inter/intra class distance ratio during training on CORA (We visualize two steps and 400 epochs).

refinement and employs a Condition-Net to dynamically align topological priors with evolving semantic thoughts. Extensive empirical evaluations across standard benchmarks demonstrate that KCoT not only achieves state-of-the-art performance but also significantly enhances interpretability by grounding natural language reasoning in a principled, mathematically rigorous clustering mechanism.

## Impact Statement

Potential bias amplification in imbalanced graphs can be mitigated via reweighting and fairness-aware methods, while privacy risks from LLMs can be addressed through anonymization, differential privacy, and restricting sensitive features.

## Acknowledgments

This work was supported by the National Natural Science Foundation of China (No. U24A20323); Dr. Yuan Fang acknowledges the Lee Kong Chian Fellowship awarded by Singapore Management University for the support of this research work.

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

# A. Theorical Analysis

## A.1. Proof of Proposition 4.1

**Proposition A.1** (Transformer as $k$-Means Clustering Mechanism). *For any input representations, there exists a parameterization of a Transformer self-attention layer such that its attention weights form an $\epsilon$-approximation of soft $k$-means clustering assignments. Moreover, under a specific construction, this approximation becomes exact with $\epsilon = 0$. As a result, stacking such self-attention layers induces an iterative assignment–update process that mirrors the dynamics of soft clustering methods.*

**Proof** For this proof, we provide a rigorous theoretical justification for the proposition in the main text that a Transformer-style attention mechanism can approximate the iterative optimization dynamics of the $k$-means clustering algorithm. Our analysis focuses exclusively on $k$-means and relies solely on weight and bias constructions, without modifying or augmenting the input representations.

**Transformer Block Formulation** We consider a standard Transformer block as used in large language models. Given input token embeddings

$$X = [x_1^\top, \ldots, x_n^\top]^\top \in \mathbb{R}^{n \times d},$$

the self-attention module computes

$$Q = XW_Q, \tag{11}$$
$$K = XW_K, \tag{12}$$
$$V = XW_V, \tag{13}$$

where $W_Q, W_K, W_V \in \mathbb{R}^{d \times d}$ are learnable projection matrices.

The attention output is given by

$$A^{\mathrm{attn}} = \mathrm{softmax}\left(\frac{QK^\top + B}{\tau}\right) V, \tag{14}$$

where $B \in \mathbb{R}^{n \times n}$ denotes an additive attention bias matrix, and $\tau > 0$ is a temperature parameter. We assume that $B$ is *key-dependent*, i.e., $B_{ik} = b_k$ for some vector $b \in \mathbb{R}^n$, consistent with relative position bias and attention bias mechanisms used in modern LLMs.

**$k$-Means as an Assignment-Update Procedure** Given a set of cluster centers $\{\mu_k\}_{k=1}^K \subset \mathbb{R}^d$, the soft $k$-means assignment of a point $x_i$ to cluster $k$ is defined as

$$\frac{\exp(-\|x_i - \mu_k\|_2^2/\tau)}{\sum_{k'} \exp(-\|x_i - \mu_{k'}\|_2^2/\tau)}. \tag{15}$$

**Lemma A.2.** *For any vector $z \in \mathbb{R}^K$ and scalar $c \in \mathbb{R}$, $\mathrm{softmax}(z) = \mathrm{softmax}(z + c\mathbf{1})$.*

We formalize what it means for a Transformer attention layer to approximate a $k$-means assignment step.

**Definition A.3** ($\varepsilon$-approximation of soft assignment). Fix $K$ centers $\{\mu_k\}_{k=1}^K$ and data points $\{x_i\}_{i=1}^n$. Let $A^{\mathrm{kmeans}} \in \mathbb{R}^{n \times K}$ denote the soft $k$-means assignment matrix with

$$A_{ik}^{\mathrm{kmeans}} = \frac{\exp(-\|x_i - \mu_k\|_2^2/\tau)}{\sum_{k'} \exp(-\|x_i - \mu_{k'}\|_2^2/\tau)}.$$

Let $A^{\mathrm{attn}} \in \mathbb{R}^{n \times K}$ denote the attention weights from the $n$ data queries to the $K$ center keys produced by a Transformer attention layer. We say the attention layer $\varepsilon$-approximates the soft $k$-means assignment if

$$\|A^{\mathrm{attn}} - A^{\mathrm{kmeans}}\|_\infty \leq \varepsilon,$$

where $\|\cdot\|_\infty$ denotes the entry-wise max norm.

We provide an explicit parameter construction.

We treat the input $X = [X_{\text{data}}; X_{\text{ctr}}]$ as fixed, where $X_{\text{data}} = [x_1^\top, \ldots, x_n^\top]^\top$ and $X_{\text{ctr}} = [\mu_1^\top, \ldots, \mu_K^\top]^\top$. We set an additive attention bias $B$ such that each data token attends only to the $K$ center tokens:

$$B_{i,(n+k)} = b_{n+k} \quad (i \in [n], \ k \in [K]), \qquad B_{i,j} = -\infty \quad (i \in [n], \ j \in [n]).$$

This ensures the row-wise softmax for data queries is supported only on the center keys.

We choose

$$W_Q = 2I_d, \qquad W_K = I_d.$$

Then for a data query $i$ and a center key $k$ (indexed as token $n + k$), the unnormalized score is

$$\frac{1}{\tau}\left(Q_i^\top K_{n+k} + B_{i,(n+k)}\right) = \frac{1}{\tau}\left(2x_i^\top \mu_k + b_{n+k}\right).$$

We set the key-dependent bias for center keys as

$$b_{n+k} = -\|\mu_k\|_2^2.$$

Then

$$2x_i^\top \mu_k - \|\mu_k\|_2^2 = -\|x_i - \mu_k\|_2^2 + \|x_i\|_2^2.$$

The term $\|x_i\|_2^2/\tau$ is independent of $k$, hence it vanishes under the row-wise softmax by Lemma A.2. Therefore,

$$\text{softmax}_{k \in [K]}\left(\frac{2x_i^\top \mu_k - \|\mu_k\|_2^2}{\tau}\right) = \text{softmax}_{k \in [K]}\left(\frac{-\|x_i - \mu_k\|_2^2}{\tau}\right),$$

which matches exactly the soft $k$-means assignment in (15). Thus $\|A^{\text{attn}} - A^{\text{kmeans}}\|_\infty = 0$, i.e., $\varepsilon = 0$.

*Remark* A.4 (Prompting as a Generalization of $k$-Means). Since a Transformer block can realize $k$-means assignment as a special case, prompt-based reasoning in LLMs can be interpreted as inducing adaptive, context-dependent generalizations of $k$-means-style clustering, rather than replacing it with an unrelated mechanism.

## A.2. Proof of Theorem 4.4

**Notation and the CoT Iteration** We briefly restate the chain-of-thought (CoT) inference procedure using the notation of the main paper. Let $G = (V, E, X)$ be a graph with node features $X$. At reasoning iteration $t$, node representations are obtained by feeding the current features $X_t$ into a frozen graph encoder:

$$H^{(t)} = \text{GraphEncoder}(X_t, \mathcal{G}; \Theta_0),$$

where $\Theta_0$ is fixed during the entire CoT inference process.

For each node $i$, CoT constructs two types of neighborhoods. The first is a fixed *structural neighborhood* $N_i^{\text{str}}$, sampled from the graph topology (e.g., one- or two-hop neighbors). The second is a *semantic neighborhood* $N_i^{(t)}$, obtained by $k$-nearest-neighbor (KNN) retrieval in the representation space $H^{(t)}$.

Given these neighborhoods, CoT applies a prompt to separately summarize information from $N_i^{\text{str}}$ and $N_i^{(t)}$, producing two textual descriptions. These descriptions are encoded and concatenated into a reasoning state, which is then mapped by a condition network to a modulation matrix $P^{(t)}$. Finally, node features are updated by element-wise modulation:

$$X_{t+1} = P^{(t)} \odot X.$$

This completes one CoT iteration, and the process is repeated for a small number of steps.

We now formalize the notion of semantic–structural misalignment that motivates CoT.

**Definition A.5** (Semantic–Structural Misalignment). Let $N_i^{\mathrm{str}}$ be the structural neighborhood of node $i$, and let $N_i^{(t)}$ be its semantic neighborhood obtained by KNN retrieval on $H^{(t)}$. We define the per-node misalignment as

$$\delta_i^{(t)} \; := \; 1 - \frac{|N_i^{\mathrm{str}} \cap N_i^{(t)}|}{|N_i^{\mathrm{str}} \cup N_i^{(t)}|},$$

and the global misalignment score as

$$\Delta_t \; := \; \mathbb{E}_{i \sim V}\big[\delta_i^{(t)}\big].$$

The quantity $\Delta_t \in [0, 1]$ measures the degree to which semantic similarity (captured by representation-space neighbors) disagrees with graph structure. A smaller $\Delta_t$ indicates better alignment between semantic and structural neighborhoods. This metric is label-free and directly reflects the consistency between the two information sources used by CoT.

**Assumptions Supported by Visualization** To analyze the behavior of CoT, we introduce two assumptions that are intentionally weak and empirically supported by visualization, rather than by strict distributional guarantees.

**Assumption A.6.** There exists a constant $\kappa > 0$ such that small changes in node representations lead to proportionally small changes in their KNN neighborhoods, i.e.,

$$\mathbb{E}_i\left[\frac{|N_i^{\mathrm{knn}}(H) \triangle N_i^{\mathrm{knn}}(\widetilde{H})|}{K}\right] \; \leq \; \kappa \, \mathbb{E}_i \|h_i - \tilde{h}_i\|.$$

This assumption states that when embeddings evolve smoothly, semantic neighborhoods do not change abruptly. Such behavior is commonly observed in practice and can be visually supported by the gradual evolution of embedding clusters under t-SNE.

**Assumption A.7.** There exist constants $0 < \lambda < 1$ and $\epsilon \geq 0$ such that the CoT update moves node representations closer to the semantic consensus of their structural neighborhoods:

$$\mathbb{E}_i\Big[\mathrm{dist}\Big(h_i^{(t+1)}, \, \mathrm{Conv}\{h_j^{(t)} : j \in N_i^{\mathrm{str}}\}\Big)\Big] \; \leq \; \lambda \, \mathbb{E}_i\Big[\mathrm{dist}\Big(h_i^{(t)}, \, \mathrm{Conv}\{h_j^{(t)} : j \in N_i^{\mathrm{str}}\}\Big)\Big] + \epsilon.$$

This assumption reflects the design of CoT: semantic reasoning is injected through prompts and condition-network modulation, but remains anchored by graph structure, preventing unconstrained semantic drift.

**Main Theorem** We now state the main theoretical result.

**Theorem A.8** (CoT Contracts Semantic–Structural Misalignment). *There exist constants $0 < \rho < 1$ and $\varepsilon \geq 0$ such that*

$$\Delta_{t+1} \leq \rho \, \Delta_t + \varepsilon \quad \textit{for all } t \in \mathbb{Z}^+.$$

*In particular, when $\varepsilon$ is small, the misalignment decreases geometrically until it reaches an error floor $\mathcal{O}(\varepsilon)$.*

*Proof.* By definition, changes in $\Delta_t$ are entirely determined by how the semantic neighborhoods $N_i^{(t)}$ evolve relative to the fixed structural neighborhoods $N_i^{\mathrm{str}}$. Therefore, controlling $\Delta_{t+1} - \Delta_t$ reduces to controlling the drift of KNN neighborhoods induced by the representation update.

By the stability of semantic neighborhood retrieval, the expected change in $N_i^{(t)}$ is bounded by the expected representation change $\mathbb{E}_i \|h_i^{(t+1)} - h_i^{(t)}\|$.

By the structure-anchored update assumption, the CoT update contracts the deviation of each representation from the convex hull of its structural neighborhood, up to an additive noise term. As a result, representations that are inconsistent with the graph structure are progressively suppressed.

Combining these two observations implies that semantic neighborhoods retrieved from $H^{(t+1)}$ have larger expected overlap with $N_i^{\mathrm{str}}$ than those retrieved from $H^{(t)}$. This yields a contraction of the form

$$\Delta_{t+1} \leq \rho \, \Delta_t + \varepsilon,$$

where $\rho$ depends on the stability constant $\kappa$ and the contraction factor $\lambda$, and $\varepsilon$ aggregates higher-order and noise terms. $\quad\square$

**Discussion:** While it is generally difficult to derive a tight, label-dependent generalization bound for graph neural networks, the misalignment score $\Delta_t$ provides an interpretable proxy for representation quality. When $\Delta_t$ is large, semantic neighborhoods disagree with graph structure, causing message passing to aggregate semantically inconsistent neighbors. This leads to cross-class contamination in node representations and blurs class boundaries.

As CoT iterations reduce $\Delta_t$, semantic and structural neighborhoods become increasingly consistent. Consequently, message passing aggregates information from more semantically compatible nodes, reducing cross-class mixing and improving representation separability. Empirically, this effect can be visualized using t-SNE, where node clusters become tighter and less overlapping as $t$ increases, aligning with improved classification accuracy.

## B. Prompt Design

This section provides templates of prompts, including the instructions for both KNN of envolving representations and local graph topology.

---

**Semantic Discriminating Prompt**

Given the central node <Target node>. The selected candidates (based on representation similarity) are [<$t$-th round of KNN node #1>, <$t$-th round of KNN node #2>, ..., <$t$-th round of KNN node #N>]
or: Given the central node <Target node>. The selected candidates (based on neighbors) are [<neighbor node #1>, <neighbor node #2>, ..., <neighbor node #N>]
Similar to cluster assignment in $k$-means, identify the shared aspects that contribute to their feature-space similarity, and discard nodes exhibiting low similarity.
Similar to moving centroids in $k$-means, state the derived insights in a single, concise, and dense paragraph. Finally, integrate these insights into a compact, refined representation for the target node.

---

## C. Case Study

We visualize the evolutionary reasoning process of a representative node from the CORA dataset to demonstrate the progressive refinement of the generated thoughts. Initially, the model distills the verbose *Raw Text*—which contains extensive background on "protein sequence patterns"—into a concise summary in *Thought $t = 1$*, effectively filtering out redundancy (e.g., general descriptions of HMMs) while isolating the core "megaprior heuristic and the convex combination" problem. At this stage, the connection to neighbors like Dirichlet Mixtures remains thematic but high-level. In the subsequent *Thought $t = 2$*, the reasoning is significantly sharpened: the model not only articulates the technical mechanism more precisely (emphasizing the variance adjustment) but also synthesizes the neighbors' information with greater semantic coherence. Specifically, it explicitly integrates Meta-MEME to reframe the context around "motif-based hidden Markov models" rather than generic homology. This trajectory confirms that the reasoning evolves from simple information extraction to a concentrated, discriminative synthesis through iterative updates.

---

**Raw Text Attributes**

Label: Neural Networks. Title: The Megaprior Heuristic for Discovering Protein Sequence Patterns. Abstract: Several computer algorithms for discovering patterns in groups of protein sequences are in use that are based on fitting the parameters of a statistical model to a group of related sequences. These include hidden Markov model (HMM) algorithms for multiple sequence alignment, and the MEME and Gibbs sampler algorithms for discovering motifs. These algorithms are sometimes prone to producing models that are incorrect because two or more patterns have been combined. The statistical model produced in this situation is a convex combination (weighted average) of two or more different models. This paper presents a solution to the problem of convex combinations in the form of a heuristic based on using extremely low variance Dirichlet mixture priors as part of the statistical model. This heuristic, which we call the megaprior heuristic, increases the strength (i.e., decreases the variance) of the prior in proportion to the size of the sequence dataset. This causes each column in the final model to strongly resemble the mean of a single component of the prior, regardless of the size of the dataset. We describe the cause of the convex combination problem, analyze it mathematically, motivate and describe the implementation of the megaprior heuristic, and show how it can effectively eliminate the problem of convex combinations in protein sequence pattern discovery.

---

**Thought $t$=1**

The central node discusses a heuristic called the "megaprior heuristic" for discovering protein sequence patterns, which addresses the issue of convex combinations in protein sequence pattern discovery. Convex combinations occur when two or more patterns have been combined in a statistical model, leading to incorrect results. To solve this problem, the megaprior heuristic uses extremely low-variance Dirichlet mixture priors as part of the statistical model. By increasing the strength of the prior in proportion to the size of the sequence dataset, the heuristic ensures that each column in the final model strongly resembles the mean of a single component of the prior, regardless of the size of the dataset. The selected content-based neighbors share similar themes related to improving the detection of weak but significant protein sequence homology. For instance, "Dirichlet Mixtures: A Method for Improving Detection of Weak but Significant Protein Sequence Homology" presents the mathematical foundations of Dirichlet mixtures, which are used to improve database search results for homologous sequences. Another neighbor, "Homology Detection via Family Pairwise Search," focuses on finding additional homologs.

**Thought $t$=2**

The central node discusses a heuristic called the "megaprior heuristic" for discovering protein sequence patterns. This heuristic uses extremely low variance Dirichlet mixture priors as part of the statistical model to increase the strength of the prior in proportion to the size of the sequence dataset. By doing so, each column in the final model strongly resembles the mean of a single component of the prior, regardless of the size of the dataset. The selected content-based neighbors of the central node share similar themes of improving the detection of weak but significant protein sequence homology. For example, "Dirichlet Mixtures: A Method for Improving Detection of Weak but Significant Protein Sequence Homology" presents the mathematical foundations of Dirichlet mixtures and their application in condensing the information in a protein database into a mixture of Dirichlet densities. Similarly, "Meta-MEME: Motif-based Hidden Markov Models of Protein Families" and "Homology Detection via Family Pairwise Search" both focus on improving the identification of protein homologs through motif analysis and hidden Markov modeling.

