# OpenReview forum: "Clustering as Reasoning: A $k$-Means Interpretation of Chain-of-Thought Graph Learning"
_ICML.cc/2026/Conference — ICML 2026 regular_

### Official Review · Reviewer_AgcH · 2026-02-28

**Soundness:** 3
**Presentation:** 3
**Significance:** 2
**Originality:** 2
**Overall Recommendation:** 4
**Confidence:** 4

**Summary:**

The paper introduces KCOT, a unified framework that reframes Chain-of-Thought (CoT) graph learning through the principle of clustering as reasoning to address the limitations of disjoint architectures and poor interpretability in existing methods. A key contribution is the theoretical proof that a Transformer block is functionally equivalent to the k-means algorithm, allowing iterative reasoning to be interpreted as structured "Assignment" and "Update" steps.

**Compliance With Llm Reviewing Policy:**

Affirmed.

**Final Justification:**

Most of my concerns have been addressed. I will update my score to 4(weak accept).

**Key Questions For Authors:**

Q1.In Proposition 4.1, you establish a formal mathematical isomorphism between a Transformer block and the k-means algorithm, yet the practical implementation relies on a pre-trained Vicuna-7B which was not explicitly optimized for such a geometric objective. Could you provide empirical evidence, such as attention map visualizations or weight distribution analyses, demonstrating that the LLM actually adopts the specific parameterization (e.g., $W_Q = 2I_d, W_K = I_d$ and specific attention biases) required for this $\epsilon$-approximation? Without such evidence, it remains unclear whether the model’s success is a direct result of the proposed theoretical alignment or simply a byproduct of the high-capacity semantic processing inherent in large-scale pre-training.

Q2.While KCOT is framed as an iterative reasoning process that mirrors the refinement of latent centroids, your sensitivity analysis in Figure 4 shows that performance peaks at $t=2$ and degrades as more reasoning steps are added. If this framework truly represents a paradigm shift toward "clustering as reasoning," one would expect additional iterations to further stabilize semantic clusters. Can you clarify why the "reasoning" process fails to converge or improve beyond two steps, and how this behavior supports your claim of a paradigm shift rather than a standard reasoning-enhanced aggregation that suffers from depth-related degradation?

Q3.Your implementation utilizes a Semantic Discriminating Prompt to filter neighbors and a Condition-Net to modulate features, which closely resembles a reasoning-enhanced aggregation strategy. Could you explicitly define the theoretical or empirical boundary where KCOT’s "clustering-based" update exceeds the expressive capabilities of a high-capacity Graph Attention Network (GAT) equipped with a semantic filter? Specifically, please justify whether the element-wise product modulation ($X_{t+1}=P^{(t)}\odot X$) and the use of semantic centroids offer a unique geometric advantage that cannot be replicated by existing attention-based neighborhood weighting and iterative feature refinement schemes.

**Limitations:**

As a tool for enhancing graph learning performance, this technique may be applied to domains such as recommendation systems, social network analysis, or financial fraud detection. The authors should discuss potential algorithmic bias (e.g., the amplification of existing biases in imbalanced graph data) as well as privacy leakage risks (e.g., LLMs inferring sensitive relationships between nodes), and propose corresponding mitigation strategies.

**Strengths And Weaknesses:**

## Strength

- A unified, mechanistic view of Graph CoT is developed by casting multi-step reasoning as a k-means–like assignment–update procedure. The Semantic Discriminating Prompt and Condition-Net follow this formulation and connect naturally to the semantic–structural alignment story, yielding a framework that is more interpretable and internally consistent than many heuristic CoT-style graph approaches.

## Weaknesses

1. The experimental comparisons rely largely on LLM-based baselines from 2023–2024, while more recent 2025-era approaches discussed in related work are not systematically evaluated. Without these stronger and newer references, the extent to which the improvements reflect a genuine state-of-the-art advance remains difficult to judge.

2. The method still follows a “neighbor selection + aggregation + update” pattern, and the Condition-Net produces an element-wise feature modulation that resembles attention weighting or gated aggregation in form. A clear expressivity gap or a capability that cannot be matched by a GAT/MPNN with a learnable semantic filter is not convincingly established, leaving the paradigm-shift framing under-supported.

3.  Interpreting reasoning as iterative clustering suggests that additional steps should improve or at least stabilize representations, yet performance peaks around two reasoning steps and then degrades. This behavior is consistent with error accumulation or over-smoothing, but the paper provides limited mechanistic analysis and few mitigation strategies (e.g., adaptive stopping, confidence-based gating), weakening the connection between the narrative and observed results.

---

> ### Author Rebuttal · Authors · 2026-03-31
>
> **1. (W1)** More recent 2025-era approaches discussed in related work are not systematically evaluated.
>
> **A1**. To address this concern, we now include the recent RGLM [1] (WWW 2026), which uses the same experimental settings as ours, and add the Reddit dataset. RGLM addresses the text-dominant bias and suboptimal graph-text alignment in existing TAG methods. The node classification results are as follows:
>
> | Method|Cora| Pubmed|OGBN-Arxiv|Reddit|
> | :-- | :--: | :--: | :--: | :--: |
> | RGLM-Decoder     | 89.85| 91.15| 75.00| 68.64|
> | RGLM-Similarizer | 89.67| 91.61| 75.14| 66.79|
> | RGLM-Denoiser    | 90.22| 90.95| 75.10| 67.64|
> | **KCoT** | **90.63** | **95.87** | **79.25** | **71.44** |
>
> Our method demonstrates clear advantages across all cases. We will include this baseline in the final version and are open to incorporating any additional suggested baselines.
>
> **2. (W2&Q3)** Could you define the theoretical or empirical boundary where KCOT’s "clustering-based" update exceeds the expressive capabilities of a high-capacity GAT equipped with a semantic filter?
>
> **A2.** The key geometric advantages of KCoT are: (1) the LLM can effectively leverage textual information to produce more discriminative embeddings, whereas GAT relies on simple, static dot-product similarity; (2) unlike GAT with a semantic filter, which uses fixed graph tokens, KCoT enables LLM and GNN to **co-evolve**, explicitly reducing semantic–structural misalignment over successive iterations, resulting in superior expressive capability.
>
> We implemented a variant using a fine-tunable BERT for text encoding followed by a GAT for aggregation, denoted as “KCoT w GAT”. The node classification results are as follows:
>
> | Method | Cora | Pubmed | OGBN-Arxiv|OGBN-Products|
> | :--- | :---: | :---: | :---: | :---: |
> | KCoT w GAT|89.13| 88.42 |74.52|83.77 |
> | KCoT | **90.63**|**95.87**|**79.25**|**86.39**|
>
> It can be seen that our method demonstrates clear advantages across all cases.
>
> **3. (W3&Q2)** Can you clarify why the "reasoning" process fails to converge or improve beyond two steps, and how this behavior supports your claim?
>
> **A3**. Thank you for this important question. We do not interpret the peak at $t=2$ in Fig. 4 as a failure of convergence. In learned iterative methods, strong performance often emerges with only a few steps: once the model has learned an effective update direction, additional iterations provide limited benefit [2-4]. LLM pretraining provides a strong prior for semantic grouping, enabling the Assignment–Update refinement to stabilize in just 2–4 steps.
>
> From this perspective, the early peak reflects **fast convergence** rather than undermining the clustering-as-reasoning framework. Once semantically consistent neighbors are identified and centroids formed (i.e., converged), further iterations could introduce noise and can slightly degrade accuracy due to over-smoothing.
>
> We will clarify this in the revision, emphasizing that KCOT implements a clustering-inspired iterative refinement that is most effective with a small number of learned steps. Fig. 6 further supports this view, showing representations progressively becoming more separable and exhibiting clustering-like dynamics during training.
>
>
> **4. (Q1)** Could you provide empirical evidence, such as attention map visualizations or weight distribution analyses, demonstrating that the LLM’s parameterization?
>
> **A4**. We thank the reviewer for this insightful question. To provide empirical evidence, we visualize the attention maps of Vicuna-7B on Cora, alongside clustering results obtained via k-means. Specifically, we compute cluster assignments $C$ and visualize $CC^T$ for comparison. For clarity, nodes are sorted by labels. The results are available at: https://anonymous.4open.science/r/18969-Figure-8F84.
>
> We observe that the attention patterns exhibit structures highly consistent with the clustering assignments, indicating similar grouping behavior. While the LLM is not explicitly parameterized as in Proposition 4.1, these results provide empirical support that it exhibits clustering-like attention dynamics aligned with our theoretical framework, rather than purely reflecting generic high-capacity semantic processing.
>
> **5**. Limitations.
>
> **A5**. We will add a discussion on ethical considerations. Potential bias amplification in imbalanced graphs can be mitigated via reweighting and fairness-aware methods, while privacy risks from LLMs can be addressed through anonymization, differential privacy, and restricting sensitive features.
>
> [1] "Toward Graph-Tokenizing Large Language Models with Reconstructive Graph Instruction Tuning." WWW 2026.
>
> [2] "Learning to optimize: A primer and a benchmark." JMLR 23.189 (2022): 1-59.
>
> [3] "Theoretical linear convergence of unfolded ISTA and its practical weights and thresholds." NeurIPS 2018.
>
> [4] "Pdhg-unrolled learning-to-optimize method for large-scale linear programming." ICML 2024.

---

> > ### Author Rebuttal · Reviewer_AgcH · 2026-04-03
> >
> > Although comparison with only one more baseline is not persuasive enough, I am satisfied with other empirical evidence provided. I will raise my score.

---

> > > ### Author Response · Authors · 2026-04-03
> > >
> > > We sincerely thank you for your positive evaluation of our work and for your thoughtful feedback throughout the rebuttal process. We truly appreciate your recognition of our empirical results and your willingness to raise the score.
> > >
> > > In response to your concern regarding the limited number of additional baselines, we have further strengthened our evaluation by incorporating two more representative methods under the same experimental settings. Specifically, GraphICL (Sun et al., NAACL 2025), which leverages structured prompt design for graph tasks, and NPC-TAG (Varolgunes et al., ICDMW 2025), which focuses on LLM–GNN alignment, are included for a more comprehensive comparison.
> > >
> > > The updated results are summarized below:
> > >
> > > | Method   | Cora      | PubMed    | ogbn-arxiv | ogbn-products |
> > > | -------- | --------- | --------- | ---------- | ------------- |
> > > | GraphICL | 83.58     | 93.18     | 73.68      | 81.48         |
> > > | NPC-TAG  | 88.60     | 95.54     | 77.04      | 81.13         |
> > > | KCoT     | **90.63** | **95.87** | **79.25**  | **86.39**     |
> > >
> > > [1] Sun, Yuanfu, et al. "Graphicl: Unlocking graph learning potential in LLMs through structured prompt design." Findings of NAACL 2025.
> > >
> > > [2] Varolgunes, Uras, Mengnan Du, and Dantong Yu. "NPC-TAG: Node Prompts for Classification on Text Attributed Graphs with LLMs." 2025 IEEE International Conference on Data Mining Workshops (ICDMW). IEEE, 2025.
> > >
> > > As shown, KCoT consistently achieves the best performance across all datasets, further supporting the robustness and effectiveness of our approach under a broader set of competitive baselines.
> > >
> > > We will incorporate these additional comparisons and corresponding discussions into the final version to further strengthen the empirical validation.
> > >
> > > | -------------------------------------------------------------------------------------- |
> > >
> > > As the discussion phase approaches its conclusion, we would like to follow up on our previous response and the additional experimental results provided.
> > >
> > > To address your concerns regarding the comprehensiveness of our evaluation, we incorporated two additional state-of-the-art 2025 baselines (GraphICL and NPC-TAG) during the rebuttal. Combined with our earlier updates, we have now added a total of three recent baselines, providing a more robust empirical validation of our proposed method.
> > >
> > > We are sincerely grateful for your previous feedback ("I am satisfied with other empirical evidence provided. I will raise my score"). We hope that these new results have fully addressed your remaining concerns regarding our experimental setup.
> > >
> > > If there are any further points you would like us to clarify, we are happy to provide more information. Otherwise, we would deeply appreciate it if you could kindly update the official score to reflect your satisfied assessment.

---

### Official Review · Reviewer_eYK2 · 2026-03-11

**Soundness:** 3
**Presentation:** 3
**Significance:** 3
**Originality:** 4
**Overall Recommendation:** 5
**Confidence:** 4

**Summary:**

This paper proposes KCOT, a framework that reframes Chain-of-Thought reasoning on graphs as a k-means clustering process. It provides a theoretical connection between Transformer blocks and k-means. It designs a Semantic Discriminating Prompt that disentangles assignment and update as distinct reasoning steps, and introduces a Condition-Net module for aligning semantic thoughts with graph topology.

**Compliance With Llm Reviewing Policy:**

Affirmed.

**Final Justification:**

After reading the response and other comments, all my concerns have been addressed. I will maintain my positive score.

**Key Questions For Authors:**

Q1. What specific factors contribute to the floor $\epsilon$, and can it be minimized?

Q2. How about the performance compared to using k-means to construct the thought?

**Limitations:**

Yes.

**Strengths And Weaknesses:**

**Strengths**

S1. Strong, non-obvious theoretical contributions bridging Transformer attention and clustering, rigorously justified. To my knowledge, this is novel.

S2. The paper is clearly organized, the mathematical derivations are readable and mostly correct, and case studies are included.

S3. Addressing a highly topical and significant challenge in the current research landscape. This paper investigates the interpretability of LLM reasoning on TAGs.

**Weaknesses**

W1. Theorem 4.4 establishes a contraction property for semantic-structural misalignment $\Delta_t$. However, the error floor $\epsilon$ is not explicitly defined in terms of graph properties. It remains unclear what specific factors contribute to this floor, and how it can be minimized.

W2. Although citations are provided to support that LLM-based grouping can outperform k-means, authors should include a direct ablation study where k-means is applied to the same text embeddings to generate thoughts. This would help isolate the actual contribution of the reasoning module.

W3. Some relevant references appear to be missing. "Towards Interpretable and Inference-Optimal COT Reasoning with Sparse Autoencoder-Guided Generation" (2025) is directly relevant for integrating clustering in CoT reasoning over graph data. It should be cited in the related work.

---

> ### Author Rebuttal · Authors · 2026-03-31
>
> **W1**. It remains unclear what specific factors contribute to this floor, and how it can be minimized.
>
> **A1**. The error floor $\epsilon$ is influenced by factors including the precision of KNN-based semantic neighborhood retrieval, the representational capacity of the frozen graph encoder, and the quality of the initial text attributes. To reduce $\epsilon$, one could use a more expressive text encoder or a stronger graph pre-training task, improving the initial latent space to better capture the underlying structural prior.
>
>
> **W2**. Although citations are provided to support that LLM-based grouping can outperform k-means, authors should include a direct ablation study where k-means is applied to the same text embeddings to generate thoughts.
>
> **A2**. We conducted an ablation where thoughts are replaced with the centroids from k-means applied to BERT embeddings, denoted as “KCoT w KMeans”::
>
> | Method | Cora | Pubmed | OGBN-Arxiv|OGBN-Products|
> | :--- | :---: | :---: | :---: | :---: |
> | KCoT w Kmeans|87.54| 90.01 |73.46|82.25 |
> | KCoT | **90.63**|**95.87**|**79.25**|**86.39**|
>
> Our method outperforms k-means in all cases, demonstrating that the LLM’s reasoning module effectively leverages text information to generate superior thoughts.
>
> **W3**. Some relevant references appear to be missing.
>
> **A3**. Thank you for the suggestion. We will cite and discuss it in the final version It applies clustering to construct the graph while we explain CoT through clustering.

---

> > ### Author Rebuttal · Reviewer_eYK2 · 2026-04-02
> >
> > I'm happy with the responses. I confirm my score for the acceptance.

---

> > > ### Author Response · Authors · 2026-04-02
> > >
> > > We sincerely thank you for your positive evaluation of our work and for your thoughtful feedback during the rebuttal process. We greatly appreciate your recognition and are committed to carefully incorporating your constructive suggestions into the final version of the paper.

---

### Official Review · Reviewer_x6qh · 2026-03-12

**Soundness:** 2
**Presentation:** 2
**Significance:** 2
**Originality:** 2
**Overall Recommendation:** 4
**Confidence:** 4

**Summary:**

This paper studies chain-of-thought (CoT) graph learning on TAGs. The key idea is to interpret CoT as a k-means-style assignment–update process, and use this view to couple LLM-based semantic reasoning with GNN-based structural propagation. Based on this idea, the paper designs a semantic discriminating prompt, a dual-neighbor thought construction scheme, and a condition-net. Experiments on node classification and link prediction over several TAG benchmarks show consistent improvements over prior baselines.

**Compliance With Llm Reviewing Policy:**

Affirmed.

**Final Justification:**

The authors’ response clarifies the novelty of the LLM-GNN co-evolution and addresses my concerns about theoretical grounding. I therefore raise my score to weak accept.

**Key Questions For Authors:**

No additional questions, all concerns are detailed in the weaknesses section.

**Limitations:**

yes.

**Strengths And Weaknesses:**

Strengths
1. The method is clearly motivated and the overall pipeline is logically connected. The transition from the k-means interpretation to prompt design, thought construction, and feature modulation is easy to follow.
2. The empirical results are comprehensive. The method outperforms prior baselines on lots of datasets and settings, and the reported gains are not limited to a single protocol.

Weaknesses
1. The method seems largely incremental over prior work[1]. Many core designs, including the iterative reasoning pipeline and several key formulations (e.g., Eqs. 1, 2, 4, 7, and 8), are highly similar. Compared with that work, the main differences here appear to be the move from text-free graphs to TAGs and the added k-means interpretation.
2. The claimed link to LLM interpretability seems overstated. Proposition 4.1 mainly shows that Transformer self-attention can represent a soft k-means-style assignment under a specific construction. This is closer to an expressivity argument than evidence that the prompted LLM in this framework actually follows such clustering dynamics.
3. The theoretical Transformer ≈ k-means result relies on a specific attention construction, while the method is a combination of Vicuna, prompting, BERT, a condition-net, and a frozen graph encoder. The paper does not sufficiently justify that the practical model behaves consistently with the theory.
4. The complexity analysis is not fully convincing in practice. It would be helpful to report GPU memory usage and the proportion of training time spent on thought updating.

> [1]. GCoT: Chain-of-Thought Prompt Learning for Graphs, KDD-2025

---

> ### Author Rebuttal · Authors · 2026-03-31
>
> **Q1**. The method seems largely incremental over prior work[1].
>
> **A1**. We respectfully disagree with the reviewer’s assessment. While we employ standard GNN operations (Eq. 1 and 4) with GraphCL loss (Eq. 2), and a condition net and Hadamard product (Eqs. 7–8), our inputs and overall design are fundamentally different from [1]. Reference [1] focuses on prompting on text-free graphs; the only superficial similarity is the iterative use of a GNN.
>
> The key distinctions of KCoT from [1] are:
>
> (1) **Novel problem formulation**: Existing TAG methods decouple LLM and GNN steps, limiting step-by-step alignment and interpretability. KCoT is the first framework where LLM and GNN co-evolve in the CoT process.
>
> (2) **Theoretical foundation**: KCoT shows that the LLM can implement the “Assignment” and “Update” steps of k-means, reframing CoT as a principled geometric optimization process rather than a heuristic.
>
> (3) **Semantic Discriminating Prompt**: Unlike prior methods, KCoT explicitly filters nodes—discarding dissimilar nodes and highlighting shared aspects—enhancing semantic discrimination.
>
> (4) **Addressing semantic-structural misalignment**: KCoT defines a misalignment metric ($\Delta_t$) and proves that its iterative CoT process geometrically reduces misalignment, yielding more stable and coherent message passing.
>
> To our knowledge, KCoT is the first framework enabling LLM–GNN **co-evolution**, offering interpretability, provable convergence, and active semantic discrimination, providing new insights for applying LLMs in graph representation learning.
>
> **Q2**. This is closer to an expressivity argument than evidence that the prompted LLM in this framework actually follows such clustering dynamics.
>
> **A2**. We clarify that Proposition 4.1 provides **mechanism-level grounding**, showing the Transformer backbone is capable of representing the assignment–update structure of k-means. The prompt and neural modules build on this foundation to perform Assignment and Update in TAGs. This does not overstate interpretability; Fig. 6 further shows representations gradually becoming separable, supporting clustering-like refinement.
>
> **Q3**. The paper does not sufficiently justify that the practical model behaves consistently with the theory.
>
> **A3**. We agree that the practical KCoT algorithm uses prompting and neural modules. Our point is that Proposition 4.1 provides an **algorithmic foundation**, aligning the architecture with the assignment–update process of k-means, which is surported by Fig. 6. The prompt and neural modules operate on this aligned backbone to improve Assignment and Update in TAGs. This design does not conflict with the theory and is consistent with algorithm-alignment approaches like algorithm unrolling [2], where learnable modules enhance a principled iterative backbone.
>
>
> **Q4**. It would be helpful to report GPU memory usage and the proportion of training time spent on thought updating.
>
> **A4**. We report the running time (s) and GPU usage (GB) below, where “LLM” indicates the time required for using LLM once, “GPU” indicates the GPU usage, and “KCoT” for using the LLM 8 times during 2 reasoning steps. Similar to our baselines and CoT in NLP area, the proportion of training time mainly relies on LLM (thought updating), which is over 95%. It can be seen that the inference time grows roughly linearly with the number of reasoning steps, but the substantial accuracy gains make this cost reasonable. The linear increase is consistent with the well-studied Chain-of-Thought methods in NLP. Efficiency can be further improved by summarizing text before the LLM, using a faster LLM, or reducing reasoning frequency, supporting scalability to large graphs.
>
> | Methods | Pubmed | Cora |
> | :--- | :--- | :--- |
> | LLM | 947.32 | 735.49 |
> |KCoT | 7827.42 | 6054.35 |
> | GPU | 14.30 | 12.57 |
>
> [2] Monga, Vishal, Yuelong Li, and Yonina C. Eldar. "Algorithm unrolling: Interpretable, efficient deep learning for signal and image processing." IEEE Signal Processing Magazine 38.2 (2021): 18-44.”

---

> > ### Author Rebuttal · Reviewer_x6qh · 2026-04-01
> >
> > I appreciate the authors' detailed response, which clarifies the novelty of the LLM-GNN co-evolution and addresses my concerns regarding the theoretical grounding. I am satisfied with these clarifications and will raise my score.

---

> > > ### Author Response · Authors · 2026-04-02
> > >
> > > We sincerely thank you for your positive evaluation of our work and for your thoughtful feedback during the rebuttal process. We greatly appreciate your recognition and are committed to carefully incorporating your constructive suggestions into the final version of the paper. We would be happy to provide any further clarifications or additional details if needed. We truly value your support and look forward to addressing any remaining questions to ensure all concerns are fully resolved.

---

### Official Review · Reviewer_sYxU · 2026-03-13

**Soundness:** 3
**Presentation:** 3
**Significance:** 3
**Originality:** 3
**Overall Recommendation:** 4
**Confidence:** 4

**Summary:**

This paper proposes a hybrid LLM–GNN framework that uses prompts to generate reasoning over node text, integrates that reasoning with graph structure through additional modules, and trains the system to perform tasks like node classification and link prediction. The core contribution is reinterpreting CoT reasoning as a process analogous to k-means clustering, where reasoning steps correspond to iterative assignment and update operations that refine node representations. Experiments showed improved performance compared with prior graph-learning and LLM-based baselines.

**Compliance With Llm Reviewing Policy:**

Affirmed.

**Key Questions For Authors:**

1.	Can you further explain how Proposition 4.1 concretely influence the actual KCOT algorithm?
2.	In the ablation section, the experiments showed the performance gap between complete KCOT framework and removal of several components. But it still remains unclear how much of the gain comes from the GNN versus the reasoning framework.

**Limitations:**

yes

**Strengths And Weaknesses:**

*Strengths*

1.	The idea of reinterpreting CoT as k-means clustering is novel and well-motivated, and strongly supported by theoretical analysis. This offers a new view of the interpretability of CoT and could stimulate future research.
2.	The proposed framework unifies LLM-based reasoning and GNNs within a single architecture, providing a way to address the problem that semantic reasoning and graph structure were processed separately.
3.	The extensive experiment results show consistent improvements over strong baselines, suggesting the framework is effective in practice.

*Weaknesses*

1.	The framework combines many components (LLM prompting, GNN encoder, iterative reasoning, KNN neighbor search, CondNet, text encoder). This makes the method difficult to understand and reproduce, and it is not always clear which component is responsible for the performance gains. Lack of open source code also hinders the reproducibility of the paper.
2.	The theoretical claim that “transformer approximates k-means clustering” seems to be loosely connected to the actual algorithm used in the framework. From my understanding, the model still relies on prompting and neural modules.
3.	The framework repeatedly calls an LLM during iterative reasoning steps for each node, which may be computationally expensive and difficult to scale for large graphs.
4.     Experiments are only on (homophilous) text-attributed graphs where graph structure information is actually not critical. It would be good to also see results on different types of graphs.

---

> ### Author Rebuttal · Authors · 2026-03-31
>
> **1**. Many components makes the method difficult to understand and reproduce.
>
> **A1**. To better understand our method, we provide Algorithm 1:
>
> Algorithm 1 KCoT
> Input: Graph $\mathcal{G} = (\mathcal{V}, \mathcal{E}, \text{X})$, text prompt $p$, CoT steps $M$
> Output: Predicted answer $\hat{y}$
> ________________________________________
> 1.	$\text{H}^{(0)} \leftarrow \text{GraphEncoder}(\text{X}, \mathcal{G})$
> 2.	$\mathcal{T}^{str} = \text{ThoughtsConstruction}(p, \mathcal{G})$
> 3.	$T^{str} = e(\mathcal{T}^{str})$
> 4.	(/ KCoT inference steps /)
> 5.	for t = 1 to M do
> 6.	$\text{H}^{(t-1)} \leftarrow \text{GraphEncoder}(\text{X}^{(t-1)}, \mathcal{G})$
> 7.	$\text{KNN}$ on $\text{H}^{(t-1)}$
> 8.	$\mathcal{T}^{t} \leftarrow \text{ThoughtsConstruction}(p, \text{KNN})$
> 9.	$T^{(t)} = e(\mathcal{T}^{str})$
> 10.	$z^{(t)} = [T^{str}||T^{(t)}]$
> 11.	$\mathbf{P}^{(t)}$ = $\text{CondNet}(z^{(t)}; \phi)$
> 12.	$\text{X}^{(t)} \leftarrow \text{X}^{(t-1)} \odot \mathbf{P}^{(t)}$
> 13.	end for
> 14.	$\hat{y}$ = Downstream($\text{X}^{(M)}$)
> 15.	return $\hat{y}$
>
> To assess the contribution of the GNN component, we perform an ablation study in which it is replaced with MLP (denoted as KCoT w/o GNN). The results are summarized as follows:
>
> ### **Table A1: Ablation study on the effects of key components (Accuracy%)**
>
> | Variant         | Node Class. (Cora) | Node Class. (Products) | Link Pred. (Cora) | Link Pred. (Product) |
> | :- | :--: | :--: | :-----------: | :--: |
> | KCoT w/o Prompt| 87.97| 82.35| 83.47| 92.05|
> | KCoT w/o GNN| 87.45| 83.65| 84.09 | 95.13|
> | **Ours** | **90.63** | **86.39** | **88.45** |**96.70**|
>
> The results indicate that the GNN component plays a more critical role on Cora for node classification, likely due to the stronger homophily inherent in its graph structure. In contrast, for the remaining datasets such as Products, while the proposed prompt mechanism contributes more significantly to the overall performance, GNN is still essential in order to achieve the best overall performance.
>
> For better reproducibility, we provided the source code in the Supplementary Material for review purposes, and it will be released publicly upon acceptance of the paper.
>
> **2**. Can you further explain how Proposition 4.1 influence the actual KCOT algorithm?
>
> **A2**. We acknowledge that KCoT relies on prompting and neural components in practice. However, Proposition 4.1 provides a **principled algorithmic foundation** by aligning the core architecture with the assignment–update process of k-means, demonstrating that the LLM backbone can support structured reasoning rather than relying on heuristics. The prompt’s role in enhancing Assignment and Update in the TAG setting builds on this aligned backbone and does not conflict with the theory-mirroring algorithm-alignment approaches such as algorithm unrolling [1], where architectures are first grounded in iterative algorithms and then augmented with learnable modules for improved performance.
>
> **3**. It may be computationally expensive and difficult to scale for large graphs.
>
> **A3**. We report the running time (s) below, where “LLM” indicates the time required for using LLM once, and KCoT for using the LLM 8 times during 2 reasoning steps. It can be seen that inference time grows roughly linearly with the number of reasoning steps , but the substantial accuracy gains make this additional cost reasonable. The linear increase is consistent with the well-studied Chain-of-Thought methods in NLP. For scalability, we demonstrated that our method can work well on larger graphs such as Ogbn-Arxiv (169,343 nodes) and Ogbn-Products (2,449,029 nodes), as shown in Table 1. Scalability can be further improved by summarizing text before LLM input, using a faster LLM, or reducing reasoning frequency.
>
> | Methods | Pubmed | Cora |
> | :--- | :--- | :--- |
> | LLM | 947.32 | 735.49 |
> |KCoT | 7827.42 | 6054.35 |
>
> **4**. Experiments are only on (homophilous) TAGs where graph structure information is actually not critical.
>
> **A4**. We have added experiments on the Reddit dataset, a social network where each node represents a user and edges indicate interactions via replies—clearly a setting where graph structure is critical. We compare against RGLM [2], and the results are shown in the table below. Our method continues to demonstrate clear advantages, thanks to the GNN evolving within the CoT process, a key distinction from traditional TAG approaches. Table A1 also shows the importance of GNN.
>
> | Reddit | Node Class. | Link Pred. |
> | :--- | :---: | :---: |
> | RGLM-Decode |68.64|81.90|
> | RGLM-Similarize |66.79|81.83|
> | RGLM-Denoiser |67.64 |81.86|
> | KCoT |**71.44**| **84.27** |
>
> [1] Monga, Vishal, Yuelong Li, and Yonina C. Eldar. "Algorithm unrolling: Interpretable, efficient deep learning for signal and image processing." IEEE Signal Processing Magazine 38.2 (2021): 18-44.”
>
> [2] Zhang, Zhongjian, et al. "Toward Graph-Tokenizing Large Language Models with Reconstructive Graph Instruction Tuning." WWW. 2026.

---

> > ### Author Rebuttal · Reviewer_sYxU · 2026-04-02
> >
> > Thanks for the clarification. I think the efficiency is still a problem and the novelty is not critically significant.  I will maintain my initial score (weak accept).

---

> > > ### Author Response · Authors · 2026-04-03
> > >
> > > We sincerely thank you for your thoughtful follow-up and for carefully considering our clarifications. We respect your decision and appreciate your balanced assessment of our work.
> > >
> > > Regarding your remaining concerns, we would like to briefly clarify:
> > >
> > > (1) Efficiency. We agree that efficiency is an important consideration. We will expand the final version to include a more concrete discussion of optimization strategies, such as LLM distillation, adaptive reasoning steps, and lightweight inference schemes. We believe these directions can substantially reduce computational overhead while preserving performance.
> > >
> > > (2) Novelty and significance. We respectfully would like to further emphasize that the core novelty lies in structurally grounding CoT reasoning within a classical algorithmic paradigm (k-means), rather than relying on heuristic or free-form prompting. This provides a principled bridge between symbolic algorithmic structure and neural reasoning, improving interpretability and offering a more generalizable design pattern for LLM-GNN integration. We will revise the paper to more clearly articulate this perspective and its broader implications.

---

### Decision · Program_Chairs · 2026-04-30

**Decision:**

Accept (regular)

**Comment:**

This paper proposes a hybrid LLM–GNN framework that uses prompts to generate reasoning over node text, integrates that reasoning with graph structure through additional modules, and trains the system to perform tasks like node classification and link prediction. The core contribution is reinterpreting CoT reasoning as a process analogous to k-means clustering, where reasoning steps correspond to iterative assignment and update operations that refine node representations. The paper is in a good quality with all positive comments. I would recommend it as Accept.